# Interactive Semantic Map Representation for Skill-based Visual Object Navigation

## Abstract

Visual object navigation using learning methods is one of the key tasks in mobile robotics. This paper introduces a new representation of a scene semantic map formed during the embodied agent interaction with the indoor environment. It is based on a neural network method that adjusts the weights of the segmentation model with backpropagation of the predicted fusion loss values during inference on a regular (backward) or delayed (forward) image sequence. We have implemented this representation into a full-fledged navigation approach called Skill-Tron, which can select robot skills from end-to-end policies based on reinforcement learning and classic map-based planning methods. The proposed approach makes it possible to form both intermediate goals for robot exploration and the final goal for object navigation. We conducted intensive experiments with the proposed approach in the Habitat environment, which showed a significant superiority in navigation quality metrics compared to state-of-the-art approaches. The developed code and used custom datasets will be publicly available.

## 1 Introduction

Accurate visual navigation to target objects in unfamiliar environments is essential for on-board mobile robot systems. In this case, the choice of embodied agent actions can be carried out by various methods: classical modular map-based motion planning algorithms Muravyev et al. (2021), end-to-end neural network models based on images of on-board cameras and/or their segmentation (Gordon et al., 2019). Classical algorithms use separate learning modules to build a map, explore the environment, or select the final goal. End-to-end models are typically trained using reinforcement learning methods, which is valuable in conditions of incomplete information about the environment.

All visual navigation methods use a semantic representation of the surrounding scene, usually formed using various learned image segmentation models. The representation of a one-shot observed map in the form of a multi-channel segmentation mask can be used directly or to build an accumulated map representation in the form of 2D (so-called birds-eye-views) (Staroverov et al., 2023), 2.5D (Ewen et al., 2022) or 3D Yang et al. (2017) semantic maps, where each map point contains a class label of the found object. The possibility of increasing the efficiency of visual navigation methods based on semantic maps is studied in special photorealistic simulators. For indoor navigation, some of the most popular environments are Habitat (Yadav et al., 2023c), AI2-THOR (Kolve et al., 2017), OmniGibson (Li et al., 2022). For the study of outdoor navigation, there are environments Carla (Dosovitskiy et al., 2017), AirSim (Shah et al., 2017), etc. There are also universal simulators for indoor and outdoor environments, such as Isaac Sim (NVIDIA, 2021). This work considers only simulation environments for the indoor navigation to given objects.

Modern works (Gadre et al., 2022; Staroverov et al., 2023), as well as solutions of such indoor navigation competitions as the Habitat Challenge[1], show that the semantic representation of the map plays a key role in increasing the success of achieving the goal and reducing the length of the resulting trajectories. The usual approach is in which semantic segmentation results are mapped using heuristic projection algorithms and noisy information about the localization of the agent (Chaplot et al., 2020). Some approaches (Ramakrishnan et al., 2022) use different rule-based methods to filter and refine segmentation results, but usually, such approaches are not correctly linked to the algorithms for planning map trajectories. The results on the success of achieving the target objects

---

[1]https://aihabitat.org

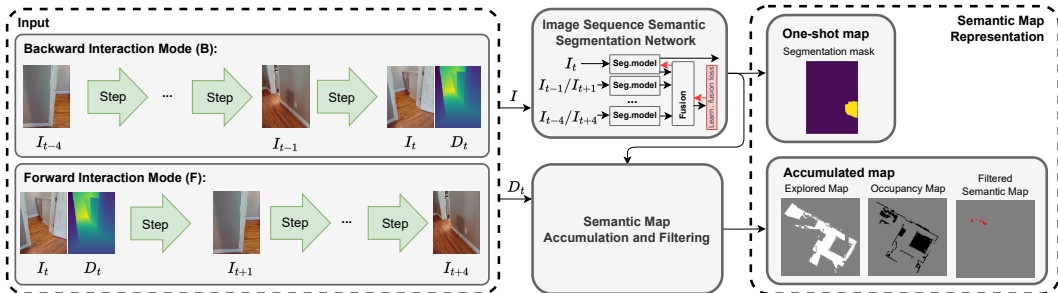

Figure 1: Proposed two-level representation of a semantic map. The one-shot map in the form of a segmentation mask is built during the interaction of the embodied agent with the environment. We consider regular (backward) or delayed (forward) image sequences as a result of such interaction. The accumulated map is a multi-channel semantic birds-eye-view map formed using an original filtering algorithm

(no more than 60% in the Habitat Challenge 2022 and 2023 competitions[2]) show that new integrated approaches to the presentation of the semantic map are needed, which would take into account the specifics of the moving embodied agent.

We propose a full-fledged navigation approach called SkillTron using adjustment of the segmentation model with backpropagation of the predicted fusion loss values during inference on a regular (backward) or delayed (forward) image sequence. SkillTron can select robot skills from end-to-end policies based on reinforcement learning and classic map-based planning methods.

Our main contributions are:

- We proposed a two-level representation of a semantic map (see Figure 1), in which a one-shot map in the form of a segmentation mask is built during the interaction of the embodied agent with the environment, and the accumulated map is formed using an original filtering algorithm for the semantic birds-eye-view map.

- We have developed a navigation approach, called SkillTron, using a proposed interactive semantic map representation with robot skill selection from end-to-end policies based on reinforcement learning and classic map-based planning methods. It allows us to form both intermediate goals for robot exploration and the final goal for object navigation.

- To study the proposed approaches, we collected several custom datasets in the Habitat Indoor Environment. This allowed us to train and test the interactive semantic segmentation model and approach for building the accumulated map. During experiments, we showed a significant superiority of the proposed SkillTron visual object navigation method in terms of quality metrics compared to state-of-the-art approaches.

## 2 RELATED WORKS

### 2.1 NAVIGATION

Like most navigational tasks, the ObjectNav task can be addressed using SLAM and deterministic planners. As a result, the agent constructs an occupancy map and a collision-free path to the goal. Despite the unknown coordinates of the goal object, methods like Frontier-based exploration (FBE) (Yamauchi, 1997) are frequently employed. A frontier is the boundary between the explored free and unexplored spaces. Frontier-based exploration essentially samples points on this frontier as goals to explore space. If the agent sees a goal type of object during this exploration, it navigates to it directly. A significant breakthrough of the learning approaches in navigation tasks was the DDPPO method (Wijmans et al., 2019), which used Proximal Policy Optimization (Schulman et al., 2017) at its core. Without mapping or planning modules, DDPPO at the PointNav task could perform 2.5 billion steps in the environment and solve the task at human-level performance. However, DDPPO

---

[2]https://aihabitat.org/challenge/2022/

demonstrated that at the ObjectNav task, the pure end-to-end RL algorithms that use vanilla visual and recurrent modules perform poorly due to overfitting and sample inefficiency. The authors of the Auxiliary task RL method (Ye et al., 2021) partially solved this by adding auxiliary learning tasks and an exploration reward during the training phase. Another promising approach to solving the ObjectNav task is to mix analytic and learned components and operate on explicit spatial maps of the environment. Such a combination of classical and learned methods was employed in the SemExp (Chaplot et al., 2020), CoW (Gadre et al., 2022), SkillFusion (Staroverov et al., 2023) methods. Usually, authors use a deterministic map module and divide a policy into a global one that, by planning on a map, outputs a short-term subgoal and a local policy that pursues that subgoal. Inspired by those works, we also built a two-level policy, but our low level consists of several independent skills and high-level switches between them depending on what the agent needs to do at a given time.

## 2.2 Interactive Computer Vision.

An embodied agent navigating through the environment can interactively update the scene's semantic representation based on new sensor data. This allows the agent to improve the semantic understanding of the scene. The recent emergence of environments for embodied agents, e.g., Habitat (Yadav et al., 2023c), AI2-THOR (Kolve et al., 2017), OmniGibson (Li et al., 2022), enabling the simulation of agent navigation and interaction with different objects, led to the development of interactive segmentation methods.

An agent can predict its future actions to improve perception quality for the next observation. In this case, the interactive segmentation would be a special case of the Next Best View selection task aiming to identify the next most informative sensor position for computer vision tasks. The recent works propose different methods to assess the informativeness of a view. The choice of the next best view can rely on the confidence score of a frozen object detector (Ding et al., 2023), segmentation quality (Yang et al., 2019; Chaudhary et al., 2023) or statistical criteria derived from the image itself (Hoseini et al., 2022). The interactive computer vision methods use both learned policies (Ding et al., 2023; Yang et al., 2019; Chaudhary et al., 2023) and predefined policies, e.g., the output of voting system (Hoseini et al., 2022), for the next best view selection. Other interactive segmentation methods aim to improve the quality of the semantic representation of an environment rather than focusing solely on image recognition. Li & Bansal (2023) improve the agent's performance on the Vision-Language Navigation Task by adding an auxiliary task to predict the view semantics for the next step. Asgharivaskasi & Atanasov (2023) propose to learn an exploration policy to decrease the uncertainty of different semantic classes by considering the motion cost.

We can highlight methods that improve the understanding of the semantics of a scene based on active exploration. The embodied agents use active exploration to facilitate the adaptation to complex and unfamiliar environments. Agents can query human expert help (Singh et al., 2022) or create pseudo-labels in testing environments using multiple points of view (Fang et al., 2020). Zurbrügg et al. (2022), Jing & Kong (2023) introduce learnable policies that consider the uncertainty of semantic maps for collecting data to fine-tune semantic segmentation models of embodied agents.

Another approach for adapting models to a test environment is fine-tuning the model during inference. Recent works Interactron (Kotar & Mottaghi, 2022) and SegmATRon (Zemskova et al., 2023) propose the use of an adaptive loss function, predicted from a frame sequence to refine object detection (Kotar & Mottaghi, 2022) and semantic segmentation (Zemskova et al., 2023). The adaptive loss function is used during both training and inference. Our work modifies the SegmATRon approach to create an interactive semantic map representation. We investigate different methods for selecting consecutive RGB observations to improve the quality of the semantic map representation.

## 3 Object Goal Navigation Task

In the context of the indoor Object Goal task as described in the literature (Batra et al., 2020), the aim is to navigate towards an instance of a specified object category $C \in \{c_1, c_2, ..., c_n\}$ (for instance, a *chair*) within an unfamiliar environment. The agent receives an observation $S = (S_{RGBD}, S_{GPS+Compass}, C)$ at each step. The action space is discrete and encompasses four types of actions: `callstop` (to terminate the episode), `forward` by $0.25m$, `turnleft`,

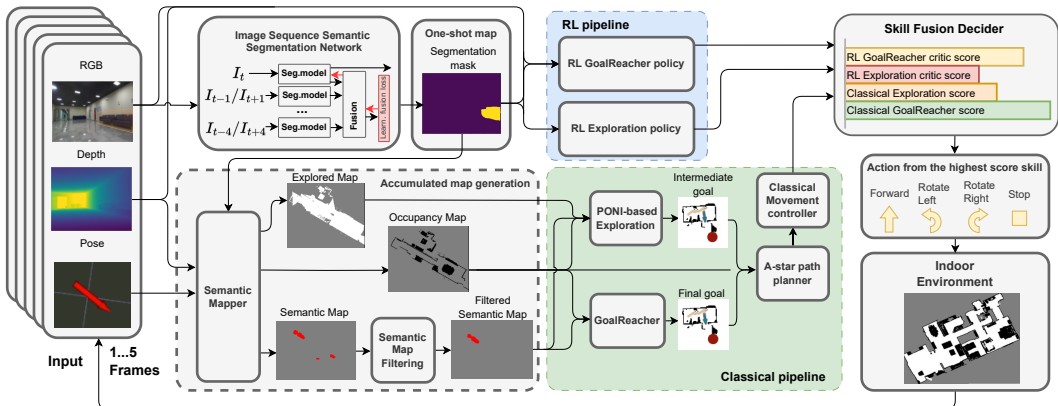

Figure 2: A scheme of the proposed SkillTRon visual navigation approach, which generates robot actions during interaction with the indoor environment. We use a fusion of robot skills, which are formed using a classic modular pipeline and end-to-end RL-based policies. Both of these pipelines utilize different layers of semantic map representation: the first uses an accumulated multi-layer map and the second works with a one-shot environment map.

and `turnright` by an angle $\alpha$ (in our experiments this angle can be equal to $30°$ or $15°$). The choice of such discrete actions is typical for indoor simulators such as Habitat (Yadav et al., 2023c).

After the agent executes the `callstop` action, the agent's performance is assessed using three primary metrics: 1) Success, where an episode is deemed successful if the agent executes the `callstop` command within 1.0m of any object of the goal type; 2) Success weighted by (the inverse normalized) Path Length (SPL), where success is weighted by the efficiency of the agent's path to the nearest object from the starting point; 3) SoftSPL, where binary success is substituted by progress toward the goal.

## 4 METHOD

### 4.1 SKILLTRON NAVIGATION APPROACH

Our proposed visual navigation approach (see Figure 2) employs two skills for solving ObjectGoal Navigation task: Exploration and Goal Reaching. Each skill is implemented with both map-based and learning-based approach. Our pipeline switches from exploration to goal reaching skill as soon as a target object is appeared in the agent's view. First, the learning-based goal reacher is used to guide the agent to the target object. After the target object is mapped, the agent reaches it using classic path planner. To avoid segmentation outliers and eliminate noises, we implement semantic map filtering with erosion, dilation, accumulation and fading.

**Map-based skills.** In our research, we utilize the Potential Function for ObjectGoal Navigation (PONI (Ramakrishnan et al., 2022)) method as a map-based exploration skill. The PONI method navigates the environment using a multi-layered map, which comprises the explored space layer, the obstacle layer, and the layers for 16 semantic classes. This map is constructed during the exploration process using depth observations and semantic segmentation masks. For exploration purposes, PONI establishes intermediate goals using a learning-based potential function on the multi-layered map. This potential function is composed of two components: the area potential and the object potential. The area potential indicates the extent of the area that can be explored from a point on the map, while the object potential represents the proximity to the goal object. The potential function neural network is based on an UNet-like map encoder (Ronneberger et al., 2015), and includes two UNet decoders for the area and object components. In the context of autonomous navigation, when a goal object is projected onto the semantic map, a path to the nearest cell of the object is constructed utilizing the A-Star path planning algorithm (Hart et al., 1968; Staroverov et al., 2023). The robot then navigates along this planned path using a straightforward path follower. The navigation episode concludes

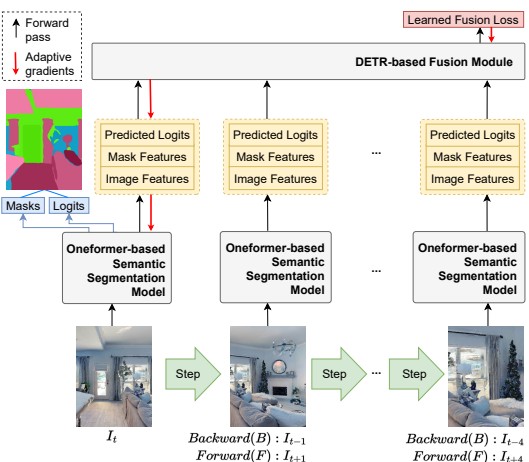

Figure 3: Image Sequence Semantic Segmentation Network: SegmATRon (B) and SegmATRon (F).

when the robot's distance to the target entity falls below a specific threshold. In our experimental setup, this threshold was set to 0.84.

**Learning-based skills.** In relation to the learning-based skills, we employ a learning-based approach within the framework of a Markov Decision Process (MDP), formally represented as $< S, A, T, R, \gamma >$. Here, $S$ represents the set of states, $A$ signifies the set of possible actions, $T(s_{t+1}|s_t, a_t)$ is the transition function, $R$ is the reward function, and $\gamma$ is the discount factor. In our proposed model, the states $s_t$ are not fully observable. The agent only has access to partial information about the state at each time step, represented as the observation $o_t$. We make an assumption that the agent is equipped with an approximator $f$ that estimates the state $s_t$ based on the history of observations: $s_t \approx f(o_t, o_{t-1}, \dots)$. In practical terms, this approximator is implemented as a component of the agent's neural network responsible for decision-making. For the training of these skills, we have employed a Decentralized Distributed Proximal Policy Optimization (DDPPO (Wijmans et al., 2019)) and a SkillFusion (Staroverov et al., 2023) approach. Its distinctiveness lies in the fact that the exploration skill is trained concurrently with the point navigation and flee tasks to develop a dynamic robust RNN encoder, and it utilizes a frozen CLIP model (Radford et al., 2021) as a robust visual image encoder. A unique feature of the Goal Reacher skill is its use of a 'not sure' action, which the agent can execute when it first identifies the goal, but upon approaching it, realizes that it was a false positive and needs to revert to exploration.

**Skill Fusion Decider.** At each step, both map-based and RL approaches are updated from the observations. And the navigational actions are taken from one of the approaches, depending from the skill fusion decider. At the exploration stage, our skill fusion decider switches between map-based and RL-based approaches comparing the RL critic score with a pre-defined threshold $\tau$. If the critic score exceeds the threshold, the agent is guided by the RL policy. Otherwise, the agent is guided by PONI. At the goal reaching stage, the skill fusion decider switches from RL-based to map-based approach as soon as the target object is mapped. If the target object disappeared from the map due to filtering, our pipeline switches from goal reaching back to exploration. Also, our pipeline switches from goal reaching to exploration if the goal object disappeared from the agent's view and is not mapped.

### 4.2 INTERACTIVE SEGMENTATION

**Transformer model.** As a baseline method for interactive image segmentation we consider the SegmATRon model (Zemskova et al., 2023) fusing the information from several frames through mechanism of a hybrid multicomponent fusion loss function. The detailed illustration of the SegmATRon is presented in Figure 3. The SegmATron model consists of two modules: a semantic segmentation model (modification of the OneFormer (Jain et al., 2023)) and a fusion module (DETR Transformer Decoder (Carion et al., 2020)). For each frame of the input image sequence the semantic segmentation model outputs image features, mask features and predicted logits. The sequence of outputs

is passed to the Fusion Module (see Figure 3). The Fusion module predicts the learned fusion loss which is used to update parameters of OneFormer. Then, the updated OneFormer makes another prediction for the first frame in the sequence. The predicted masks and logits are considered as final semantic segmentation of the first frame in the sequence.

**Adaptive Fusion Loss Function.** The adaptive fusion loss function $\mathcal{L}_{fusion}(\phi, \theta, \mathbf{F})$ is parameterized by Fusion Module parameters $\phi$ and depends on parameters $\theta$ of the OneFormer model and a sequence of frames $\mathbf{F}$. The parameters $\theta$ are updated by backpropagation through adaptive gradients. During the training process the Fusion module parameters $\phi$ and the OneFormer parameters $\theta$ are optimized jointly. The goal is to minimize multicomponent segmentation loss $\mathcal{L}_{segm}(\theta, \mathbf{F})$ over all ground-truth sequences $\mathbf{R}_{all}$.

$$\min_{\theta,\phi} \sum_{\mathbf{F} \in \mathbf{R}_{all}} \mathcal{L}_{segm}(\theta - \alpha \nabla_\theta \mathcal{L}_{fusion}(\phi, \theta, \mathbf{F}), \mathbf{F}). \tag{1}$$

The segmentation loss function is the original OneFormer loss function (Jain et al., 2023) without the contrastive loss term. Thus,

$$\mathcal{L}_{segm} = \lambda_{cls}\mathcal{L}_{cls} + \lambda_{bce}\mathcal{L}_{bce} + \lambda_{dice}\mathcal{L}_{dice}, \tag{2}$$

where, $\mathcal{L}_{cls}$ – cross-entropy loss for class prediction, binary cross-entropy ($\mathcal{L}_{bce}$) and dice loss ($\mathcal{L}_{dice}$) are controlling mask predictions.

**Navigation Image Sequences.** The SegmATRon model is trained on offline data of image sequences that were collected in a simulation environment. However, during navigation, sequences of images appear online, and the frames content depends on the actions chosen by the agent. Therefore, the SegmATRon can use a buffer storing frame sequences which is updating interactively. Two parameters controls the buffer properties: the number of images and the sequence order. In our experiments we consider buffers of different lengths: 2 images, 3 images and 5 images. These sizes correspond to 1, 2 and 4 additional frames (steps) used to refine semantic segmentation. In addition, we conduct experiments with different image sequence orders in the frame buffer. We call the image sequence order "backward" when a sequence of frames $\{I_t, I_{t-1}, ..., I_{t-n}\}$ is used for prediction at step $t$, where $n$ is the number of additional frames that the SegmATRon (B) uses. In the case of "forward" image sequence order, at time $t + n$ the SegmATRon (F) makes a prediction for frame $I_t$, using "future" frames relatively to frame $I_t$, i.e. $\{I_t, I_{t+1}, ..., I_{t+n}\}$. Thus, the semantic map is updated with a delay, but the SegmATRon (F) model can use frames in which the goal objects are better viewed. When a goal is observed, the agent navigates towards it, so the "forward" images could have more information about the goal than "backward" images.

### 4.3 SEMANTIC MAP ACCUMULATION AND FILTERING

Learning-based semantic segmentation predictions sometimes have noises, especially in far objects. Projecting these noises onto the map causes semantic mapping outliers and leads to reaching a false goal and unsuccessful finish. To prevent this, we implement semantic map filtering consistting of erosion, dilation, map accumulation and fading. A scheme of semantic map filtering is shown in Figure 4.

At each step $t$, a local semantic map $L_t$ is created using pose and depth observation with the SegmATRon-predicted semantic mask. First, an erosion is applied to the local map to filter out semantic segmentation outliers. Next, to return the initial size to the target objects, dilation is applied as a convolution. After that, the local map is fused with the global map $M_{t-1}$ to obtain the updated global map $M_t$. The fusion is implemented as accumulation and fading. The global map values in the cells containing a goal object in the local map are increased by 1 (accumulation), and the global map values in the cells not containing a goal object are multiplied by a decay coefficient $\alpha < 1$ (fading). The global map values in the cells not covered by a local map are not changed. A cell is considered a target object cell if its value in the global map exceedes some pre-defined threshold $T$. A formal description of the semantic map filtering is shown below:

$$L_t = dilation_k(erosion_k(L_t)), \tag{3}$$
$$M_t = (M_{t-1} + L_t) \cdot (L_t + \alpha(1 - L_t)C_t + (1 - L_t)(1 - C_t)), \tag{4}$$

where $C_t$ is the coverage mask of the local map $L_t$.

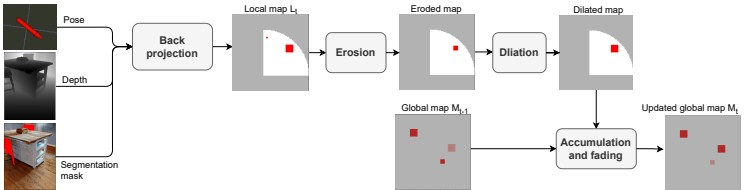

Figure 4: Scheme of the proposed semantic map filtering.

# 5 EXPERIMENTS

## 5.1 EXPERIMENTAL SETUP

**Navigation.** We validate our navigation pipeline in the Habitat environment. We use validation scenes from the Habitat Matterport3D semantics (HM3DSem) v0.2 dataset (Yadav et al., 2022) and select 108 episodes with the following distribution of goal categories: potted plant (18), chair (15), chair (25), tv (17), bed (17), toilet (16). The validation scenes were never seen during training phase. We use the environment configuration from Habitat Challenge 2023 (Yadav et al., 2023b) with slight changes. The use of adaptive gradients during inference requires more computational sources for SegmATRon method. Therefore, we increase the maximum amount of time per episode up to 1500 seconds and fix the maximum number of steps to 500. We conduct experiments on a server with 1 Nvidia Tesla V100 GPU. We repeat each validation navigation experiment 5 times and report the mean value of standard navigation metrics: Success rate, SPL and SoftSPL as they defined in Habitat Challenge 2023 (Yadav et al., 2023b).

**Semantic maps representation.** We assess the quality of semantic map representation for different methods using the same observation dataset. We collect this dataset recording observations during navigation with SkillTron approach with SegmATRon (B) (1 Step) semantic method. At each step, we save the agent's position, the tilt of its head, the depth map, GT semantics, RGB observation and SegmATRon (B) (1 Step) predictions of semantic mask. Then, we make predictions on the saved image sequence using different versions of the SegmATRon approach and various sets of hyper-parameters for semantic map construction. This approach allows us to distinguish between the navigation performance and the quality of semantic map representation. We evaluate the quality of semantic map representation using metrics of Closeness-sigmoid, IoU described in the Section 5.2.

**Interactive segmentation model training.** We train SegmATRon on offline data collected in HM3DSem v0.2 dataset (Yadav et al., 2022). We collect a training dataset in 1160 random points of train scenes and a validation dataset in 144 random points of validation scenes. We make sure to include some random viewpoints of goal categories in our datasets. The image sequences for the SegmATRon models training are obtained by considering all possible combination of 4 actions made from starting random points. We consider the following set of actions: turn left, turn right, look up, look down, and move backward. By moving backward the agent can observe a scene from more distant point of view. The tilt angle for look up and look down action is fixed to 30°. In our experiments we consider turn angles of 15°and 30°. For both values of turn angle we collect datasets starting from the same root points. Since during the training and validation process the image sequences are chosen randomly from available combinations, we use the same sequence of weights for SegmATRon (B) and SegmATRon (F) approaches.

In our experiments we use a custom mapping of 1624 original categories HM3DSem v0.2 dataset (Yadav et al., 2022) to 150 categories of ADE20k (Zhou et al., 2019). Thus, we can efficiently fine-tune our segmentation models starting from weights trained on the rich semantics of ADE20k (Zhou et al., 2019) without pseudo-labeling. We consider this mapping as ground-thruth during the training and validation process of SegmATRon. Additionally, we use this mapping to compute ground-truth semantic maps to assess the quality of semantic map representation.

**Baselines.** As a baseline method for our approach to semantic map representation we consider the SkillFusion (Staroverov et al., 2023) method - the winner of Habitat Challenge 2023 (Yadav et al., 2023b). We distinguish the role of interactive semantic map representation by considering

Table 1: Semantic map quality with respect to filtering parameters

| Decay coef. $\alpha$ | Threshold $T$ | Closeness-sigmoid | IoU | FPR | FNR |
|---|---|---|---|---|---|
| 0.8 | 1 | 0.366 | 0.365 | 0.23 | 0.02 |
| 0.9 | 1 | 0.392 | 0.358 | 0.24 | **0.01** |
| 0.8 | 2 | **0.240** | 0.371 | **0.12** | 0.03 |
| 0.9 | 2 | 0.272 | **0.395** | 0.13 | 0.02 |

Table 2: Semantic map quality with different SegmATRon approaches

| Semantic method | Number of steps | Closeness-sigmoid | IoU | FPR | FNR |
|---|---|---|---|---|---|
| SegmATRon (B) | 1 | 0.272 | 0.395 | 0.13 | 0.02 |
| SegmATRon (B) | 2 | 0.192 | 0.434 | 0.10 | **0.01** |
| SegmATRon (B) | 4 | **0.175** | **0.446** | **0.09** | 0.03 |
| SegmATRon (F) | 1 | 0.217 | **0.418** | 0.11 | 0.02 |
| SegmATRon (F) | 2 | **0.184** | 0.415 | 0.08 | 0.02 |
| SegmATRon (F) | 4 | 0.213 | 0.371 | 0.09 | 0.05 |

the Single Frame baseline for our approach: the OneFormer (Swin-L backbone) model (Jain et al., 2023) fine-tuned on the datasets collected for SegmATRon training.

## 5.2 SEMANTIC MAPS

We estimate the quality of semantic maps using two metrics: Closeness-Sigmoid and Intersection over Union (IoU). The Closeness-Sigmoid metric is calcluated as an average distance from the predicted semantic map cells to the closest cell of the ground truth semantic map, followed by a sigmoid function. The resulting metric has value range between 0 and 1. If there are no predicted map cells, the metric is considered 1. Also, we estimate False Positive Rate (FPR) and False Negative Rate (FNR) metrics. The FPR value is the part of episodes where the predicted map contains a spurious goal object, and the ground truth map contains no goal objects. The FNR value is the part of episodes where the ground-truth semantic map contains goal objects and the predicted semantic map is empty.

First, we choose the optimal filtering hyperparameters: decay coefficient $\alpha$ and threshold $T$. For this purpose, we test four pairs $(\alpha, T)$ with SegmATRon (B) (1 Step). The results of the tests are shown in Table 1. According to all the metric values, we choose $\alpha = 0.9$ and $T = 2$ for our SegmATRon experiments. Next, we test different SegmATRon approaches with the chosen filtering hyperparameters. The results of the tests are shown in the Table 2. According to the both Closeness-Sigmoid and IoU metrics, the best approach is the SegmATRon (B) with 4 steps - this approach reaches Closeness-Sigmoid 0.175 and IoU 0.446. The results with two-step SegmATRon (F) are relatively close - Closeness-Sigmoid 0.184 and IoU 0.415.

## 5.3 NAVIGATION WITH DIFFERENT SEMANTIC MAPS

**Comparison with different baselines.** The SkillTron approach significantly outperforms various state-of-the-art methods listed on the public leaderboard of Habitat Challenge 2023 Test Standard Phase (Yadav et al., 2023a), (Yadav et al., 2023b) (see Table 3). There is no public code available for the ICanFly and Host_74441_Team approaches, therefore we report the navigation metrics based on its performance on the Habitat Challenge 2023 Test Standard dataset (Yadav et al., 2023b). As one can see from Table 3, the SkillTron surpasses the state-of-the-art approach SkillFusion (Staroverov et al., 2023) by a considerable margin of 4% of success rate, 2% of SPL and 2% of SoftSPL metrics. The interactive semantic map representation plays an important role in the performance of SkillTron method. Our best interactive segmentation network uses two additional frames and backward image sequence. The SkillTron method with interactive semantic map representation shows the increase of 2% of the Success and 1% of SoftSPL metrics compared to the baseline SkillTron using OneFormer as the semantic segmentation network.

**Ablation on number of steps (Additional Frames).** Long image sequences carry more information about the environment. We vary the number of additional frames used to predict a segmentation mask. We consider the backward image sequence order for these experiments and turn angle of

Table 3: Performance of SkillTron as compared to the baselines on the HM3DSem v0.2 dataset.

| Method | Exploration | Goalreacher | Semantic method | Success | SPL | SoftSPL |
|---|---|---|---|---|---|---|
| Host_74441_Team[2] | - | - | - | 0.12 | 0.05 | 0.27 |
| ICanFly[2] | - | - | - | 0.43 | 0.26 | **0.37** |
| SkillFusion | PONI+RL | Classic+RL | OneFormer | 0.55 | 0.26 | 0.34 |
| SkillTron | PONI+RL | Classic+RL | OneFormer (finetuned) | 0.57 | **0.28** | 0.35 |
| **SkillTron** | **PONI+RL** | **Classic+RL** | **SegmATRon (B) (2 Steps)** | **0.59** | **0.28** | 0.36 |

Table 4: Ablation study. Number of steps (Additional Frames) and Turn Angle Values.

| Method | Semantic method | Number of steps | Turn Angle | Success | SPL | SoftSPL |
|---|---|---|---|---|---|---|
| SkillTron | OneFormer | - | 30° | 0.57 | 0.28 | 0.35 |
| SkillTron | SegmATRon (B) | 1 | 30° | 0.58 | **0.29** | 0.35 |
| **SkillTron** | **SegmATRon (B)** | **2** | 30° | **0.59** | 0.28 | **0.36** |
| SkillTron | SegmATRon (B) | 4 | 30° | 0.57 | 0.27 | 0.35 |
| SkillTron | SegmATRon (B) | 1 | 15° | 0.51 | 0.25 | 0.33 |
| SkillTron | SegmATRon (B) | 2 | 15° | 0.49 | 0.24 | 0.31 |
| SkillTron | SegmATRon (B) | 4 | 15° | 0.50 | 0.25 | 0.32 |

30°. The Table 4 shows the navigation metrics for SegmATRon (B) with different number of steps (additional frames). The SegmATRon (B) (2 Steps) demonstrates the increase of success rate and SoftSPL metrics comparing to the SegmATRon (B) (1 Steps). However, the SegmATRon (B) (4 Steps) shows decrease of navigation metrics comparing to the SegmATRon (B) (2 Steps). These results are in agreement with the dependence of semantic maps quality metrics on the number of steps (see Section 5.2) for 1 and 2 additional frames. The 4 additional frames may not be optimal for navigation, since the processing of longer image sequences slows the navigation pipeline.

**Ablation on the turn angle values.** The continuity of view during navigation is determined by agent's turn angle. We conduct an ablation study to investigate if a smaller turn angle would increase the performance of the Fusion module of the interactive segmentation network. We consider 15 °- a half of turn angle in the configuration of discrete action space of Habitat Challenge 2023 (Yadav et al., 2023b). We retrain learning-based skills in the new action space and the interactive segmentation network. The 15°turn angle is less effective for navigation than 30 °(see Table 4) for all considered number of steps. It can be partially since the smaller turn angle requires more steps during the exploration phase of object goal navigation.

# 6 CONCLUSION

In this paper, we proposed a new visual navigation approach called SkillTron, which uses a two-level interactive semantic map representation, as well as fusing exploration and goal reacher skills of the robot. To build a one-shot map level, we studied in detail the neural network method, which corrects the weights of the segmentation model based on the predicted values of fusion loss during inference on a regular (backward) or delayed (forward) image sequence. We showed that the backward interaction mode provided a more accurate construction of a 2D accumulated semantic map, which was then used for navigation. We demonstrated that the proposed combination of an RL-based navigation pipeline and a classic modular approach using learnable modules outperformed existing state-of-the-art approaches in indoor environments from the Habitat simulator.

As limitations, it should be noted that the selected semantic segmentation network is resource-demanding, as well as the fact that proposed visual navigation approach has only been tested in simulation environments. Further directions for the development of the proposed approaches could be the study of transferring the method of visual navigation to a real robot, the use of other more compact basic models of semantic segmentation and image sequence fusion to form a one-shot representation of a semantic map. Considering the interactive segmentation as a separate robot skill with learned action policy is also of interest for future work.

---

[2] We report metrics from public leaderboard of Habitat Challenge 2023 Test Standard Phase (Yadav et al., 2023a), (Yadav et al., 2023b) due to absence of public code implementation.

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
