# OpenReview forum: "Interactive Semantic Map Representation for Skill-based Visual Object Navigation"
_ICLR.cc/2024/Conference — Submitted to ICLR 2024_

### Official Review · Reviewer_gNN6 · 2023-10-31

**Soundness:** 2 fair
**Presentation:** 3 good
**Contribution:** 2 fair
**Rating:** 3
**Confidence:** 3

**Summary:**

The authors propose a method for object navigation. The agent is equipped with an RGBD camera and GPS+compass. The proposed approach uses a skill fusion decider that selects whether to execute steps from an RL policies or classical map-based approaches. As the agent traverses the environment, it builds a semantic map of the environment that is robust to noise, achieved through image dilation/erosion and temporal accumulation of confidence. They benchmark their approach on the Habitat 2023 object navigation challenge and show the proposed approach outperforms the challenge winner on the validation scenes.

**Strengths:**

* The authors propose an interesting approach for object navigation that outperforms the baselines. Because it uses both classical map-based approaches and learned approaches, it is able to leverage the best of both worlds.
* Approach is robust to segmentation noise
* Writing is generally clear with a few gaps here and there (see below).
* Useful and reasonable experiments.
    * The proposed approach outperforms the baselines on a recent object navigation challenge, though the improvement is marginal under the OneFormer setting.
    * Ablation study is useful. We can see how different parameter setups affect performance (e.g. number of frames, agent turn angle).

**Weaknesses:**

* The motivation for the work is not very clear. Why do we think it's necessary to have the 4 combinations: classical exploration, classical goal reaching, RL-based exploration, RL-based goal reaching? What's the benefit over having just classical or just RL?
* Writing is generally clear with a few gaps here and there, such as the definition of L_{fusion}, or when RL-based exploration is used (it is in Fig. 2 but not mentioned in Sec. 4.1 Skill Fusion Decider).
* Navigation Image Sequences in 4.2 is not very well motivated. Is it to generally increase prediction accuracy by providing more contextual information about the scene/visuals? It's not clear how this relates to the segmentation model being trained on offline data.
* Very similar to SkillFusion (Staroverov et al.). The contributions over this prior work seem minor.

**Questions:**

* What kind of sensor noise occurs at test time? Does the agent use ground truth odometry and depth?
* The proposed method is very similar to SkillFusion (Staroverov et al.). What are the main differences and contributions over this prior work?
* Please clarify the definition of L_{fusion}
* When is RL-based exploration used?
* There is a missing ablation on the proposal in 4.3. What happens when there is no erosion/dilation/accumulation/fading?

---

### Official Review · Reviewer_H7Mw · 2023-11-01

**Soundness:** 2 fair
**Presentation:** 1 poor
**Contribution:** 2 fair
**Rating:** 1
**Confidence:** 4

**Summary:**

This paper proposes SkillTron, a navigation approach that uses a newly proposed two-level representation of interactive semantic map with robot skill selection for the object goal navigation task. Experiments are conducted on Habitat Matterport3D challenge.

**Strengths:**

- The paper presents a valid and reasonable system for the ObjectGoal Navigation task.
- Experiments show some improvement over baselines.

**Weaknesses:**

- The paper writing needs significant improvement for both English and the writing flow. I cannot understand a lot of sentences. And I can barely read into understanding the paper details.
- The paper is more like an experiment report after participating the Habitat ObjectGoalNav challenge than a paper. The problem formulation is very specific for the specific challenge. I'm not sure if the paper has any generality that makes it interesting for general ICLR paper readers.
- The proposed system mostly uses previous works as its components. It's unclear if there is any technical contribution in the method.
- Experiments are weak, only comparing to 2 team submissions.

**Questions:**

see weakness

---

### Official Review · Reviewer_85u7 · 2023-11-05

**Soundness:** 1 poor
**Presentation:** 1 poor
**Contribution:** 1 poor
**Rating:** 1
**Confidence:** 4

**Summary:**

This work presents a new semantic map representation combined with leveraging a planning policy which uses multiple prior methods in a single system to explore and reach semantic targets in novel environments. The correctness of the 2D semantic map is measured as well as semantic navigation performance in novel environments in simulation.

**Strengths:**

The authors do a good job of performing ablations of their own method to determine the relative contribution of each of the components of their pipeline. The diagrams of their proposed pipeline are well made and clear. The description of the approach is also presented clearly and the motivation and details explained well.

**Weaknesses:**

The experimental results have substantial errors. The proposed method, SkillTron+SegmATRon, is evaluated on a manually selected subset of validation episodes from the HM3D dataset. Then, numbers from the Habitat Challenge are input in comparison which evaluate on a completely different set of test episodes which are not public. SkillTron+SegmATRon is not accurately compared (on one shared test set!) against published state-of-the-art methods for semantic navigation.

If the authors want to have a 1-1 comparison with alternate approaches without manually rerunning baselines on their custom testset, they should evaluate SkillTron+SegmATRon on the full test set for HM3D (the dataset they use) and then numbers from other papers which evaluate on this standard and public test set can be directly input into the comparison table. If they would like to evaluate against methods which run in the Habitat Challenge, they should submit their code to the Habitat Challenge so that they have also run on the same secret test set. However, running on the challenge should not replace running on a standard publicly available test (i.e. the HM3D test set) so that their paper has publicly reproducible evaluation results.

Also, the authors claim “significant improvement” in the experiment section of SkillTron+SegmATRon over SkillTron+OneFormer (which seem to have been evaluated on the same test set) when the change in both success rate, SPL, and SoftSPL is 0.02 or less in every case and no standard error metrics are reported. So, it is not clear in the ablations whether the semantic map method actually yields a statistically significant change in performance at all.

In addition to the experimental errors, there are many statements throughout the paper in comparisons to related work which also are conjectures stated as facts. For example, in the introduction “such approaches are not correctly linked to the algorithms for planning map trajectories” regarding the approach in (Ramakrishnan et al., 2022). Why is the RL approach used to plan a map trajectory in this paper “incorrect”?

Also, the authors should consider in the presentation of their method motivation: If the goal of their work is to find a highly accurate semantic map of the observed area, why should highly performant semantic SLAM methods not be used?

**Questions:**

Note: the correct review template is not used. The heading says “Published as a conference paper at ICLR 2024.” instead of “Under review as a conference paper at ICLR 2024”. The authors are using the camera ready template with the author names deleted - not the anonymized submission template.

---

### Official Review · Reviewer_Xvsx · 2023-11-07

**Soundness:** 2 fair
**Presentation:** 2 fair
**Contribution:** 2 fair
**Rating:** 3
**Confidence:** 3

**Summary:**

The paper addresses the problem of visual object navigation by combining active visual exploration and visual navigation with classical map exploration approaches. Regarding visual based navigation, it builds on SegmATRon, using an adaptive loss function to improve object segmentation during inference and create interactive semantic map representations. The proposed method improves the state-of-the-art (sota) on simulated data from the Habitat simulator and the Habitat challenge.

### Comments after the rebuttal
The authors did not provide answers to the comments made by me and the other reviewers. Based on these comments, there are important issues that need to be addressed before the paper can be accepted for publication.

**Strengths:**

The paper proposes a complete architecture containing both learning-based and classic map-based planning methods to improve the performance on the visual object navigation task. The two approaches are combined using a decision module. Navigation is performed either by choosing actions based on a reinforcement learning pipeline fed by semantic segmentation of the scene, or by classical map-based navigation actions achieving both exploration and goal reaching. In the case of visual based navigation, the proposed method exploits the advantages of SegmATRon to achieve more reliable segmentation of the scene via online modification of the segmentation model weights during inference by using a buffer of frames collected during the execution of the actions.

**Weaknesses:**

The writing quality of the paper and the use of English could be improved. The paper also suffers from the quite tight page limit as the descriptions both of the full model and the individual modules are quite brief without covering crucial details regarding their motivation and the relative design choices. In fact, it appears to be very hard to reproduce this work based on the content of the paper, as many details regarding the architecture and the parameters of each module are not provided.

Another crucial aspect regards relation to prior work. It appears that the proposed architecture is heavily based on the SkillFusion paper, including a more powerful semantic segmentation approach (SegmATRon). This is not clearly stated in the text. It is crucial to make a detailed comparison between the SkillFusion model and the one presented in this work (SkillTRon). In general, it seems that the proposed method is a combination of sota methods. This is not a problem in general, but a comprehensive comparison with the corresponding methods should be provided, clarifying what are the relative improvement and contributions of the proposed work with respect to previous ones.

**Questions:**

- Please describe the difference with respect to the SkillFusion model.

---

### Meta-Review · Area_Chair_mHiT · 2023-12-10

**Metareview:**

Synopsis: This paper presents an algorithm to learn and refine a semantic map representation over the course of skill-based interactions with an environment. Skills include RL-generated ones and planner-generated ones. A decision-making component chooses the final skill to execute in the world at any given time.

Strengths:
+ The approach proposes an interesting fusion of classical navigation and learning-based navigation
+ The idea of on-the-fly finetuning of semantic segmentation based on the results of skill-based interactions is promising
+ Good ablation studies

Weaknesses:
- The technical presentation has lots of room for improvement - key parts of the approach are not clear (the reviews mention specifics)
- The discussion on relation to previous work is not precise at places
- The experimental results omit key baselines. It is not clear why the authors did not submit directly to the habitat benchmark.

**Justification For Why Not Higher Score:**

There is consensus among both the reviewers, and from my read of the paper, that the paper could improve in its technical presentation, its positioning wr.t. the state of the art, and the empirical evaluation.

**Justification For Why Not Lower Score:**

N/A

---

### Decision · Program_Chairs · 2024-01-16

Reject